# Characterization of the Nozzle Ablation Rate Based on 3D Laser Scanning System

Kaining Zhang, Chunguang Wang *, Qun Li and Zhihong Wang

State Key Laboratory for Strength and Vibration of Mechanical Structures, School of Aerospace Engineering, Xian Jiaotong University, Xi'an 710049, China
* Correspondence: wangchunguang@xjtu.edu.cn

**Abstract:** The nozzle of solid rocket motor (SRM) is easily ablated by high temperature, high pressure, high speed, and corrosive particles, which affects the stability of rocket flight. Therefore, the measurement and characterization of the nozzle ablation rate are helpful in providing some guidance for the design of nozzle material and structure. However, due to the high surface roughness of the composite nozzle after ablation, it is difficult to obtain an accurate ablation rate by contact measurement methods. The 3D laser scanning system is a 3D non-contact measurement technology using structured light technology, phase measurement technology, and computer vision technology. It has the advantages of non-contact, large scanning size, flexibility, and portability. In this paper, a 3D reconstruction of the ablation nozzle is carried out based on the 3D laser scanning system. Additionally, the ablation rate of the nozzle is measured without cutting the actual specimen. Furthermore, the pressure, temperature, and surface convective heat transfer coefficient trends are numerically calculated and compared with the ablation rate. Additionally, the empirical formula between ablation rate and pressure, temperature, convective heat transfer coefficient is obtained empirically by the inversion analysis method. The empirical formula can provide theoretical guidance for nozzle size design and optimization. The results show that the non-contact 3D laser scanning system is a valuable method for reconstructing the model of the ablated nozzle, and the empirical formula of ablation rate can accurately predict the ablation rate of the nozzle.

**Keywords:** solid rocket motor; ablation rate; nozzle; 3D laser scanning system; inversion analysis method

## 1. Introduction

The nozzle is a crucial device for converting thermal energy generated by the SRM into kinetic energy. The throat insert is the smallest channel of the nozzle. The high-temperature gas generated in the combustion chamber becomes supersonic through the throat insert, and propulsion is generated as a result [1,2]. Therefore, the service condition of the nozzle is extremely harsh [3]. Under the working conditions of the SRM, the nozzle surface is damaged by the high temperature, high pressure, and high-speed particles produced by solid propellant combustion will cause nozzle ablation [4,5], resulting in the change of the nozzle surface size and the expansion of the diameter. Therefore, the efficiency of the nozzle is reduced, and finally the performance of the engine is reduced, and even the burn-through failure is caused.

Ablation is the main factor that causes the size change of the nozzle. It affects the propulsion characteristics [6], which can be divided into two main categories: thermochemical ablation and mechanical erosion [7]. Therefore, the measurement and characterization of the nozzle ablation rate are helpful in providing some guidance for the design of nozzle material and structure. However, due to the high surface roughness of the composite nozzle after ablation [8,9], it is difficult to obtain an accurate ablation rate by contact measurement methods. Therefore, many non-contact measurement methods have been developed to

obtain the ablation rate of the nozzle. The ablation rate of 3D reconstruction before and after nozzle ablation was measured and calculated using a micro X-ray system [6]. The linear accelerator industrial computed tomography (CT) was used to obtain the cross-sectional structure of the nozzle, and the throat insert morphology was characterized by a scanning electron microscope [10]. Using the scanning electron microscope (SEM), the microstructure and morphology of the throat insert Carbon/Carbon (C/C) composite were analyzed [11]. The charge-coupled device (CCD) system was used to obtain the image of the throat insert, and the diameter of the throat insert was measured [12]. However, the measurement range of these non-contact methods is too small to measure the nozzle ablation rate in the whole field.

The 3D laser scanning system is a 3D non-contact measurement technology using structured light technology, phase measurement technology, and computer vision technology. Additionally, the 3D structure point cloud data can be obtained by the triangulation and deformation of the high-resolution fringe patterns, which are taken by these cameras. The 3D laser scanning system has been widely used in the automotive industry [13], aerospace [14], archaeology [15], medical treatment [16], etc. In order to measure the ablation rate of the nozzle, a non-contact 3D laser scanning system is established in this paper. The 3D laser scanning system consists of two optical cameras, a light source, and a fringe pattern projector. The 3D structural data is obtained by projecting laser gratings onto the nozzle surface. Compared with traditional modeling methods such as industrial CT, CCD system, and scanning electron microscope (SEM), the 3D laser scanning system is more straightforward, more economical, has a larger scanning size, and can obtain a complete 3D model of the ablated nozzle. This 3D model makes it unnecessary to cut the actual specimen for the analysis.

In addition, the prediction of the nozzle ablation rate is crucial [2,17], because the ablation of the nozzle can cause a pressure drop in the combustion chamber and insufficient combustion. In fact, the production process of heat-resistant material for the nozzle is complicated, the material components are different, and the chemical and physical reactions of the material are complex. The complex application scenarios and service environments of SRM bring challenges to the accuracy of the ablation performance prediction model. The chemical reaction between the nozzle material and the oxidizer in the fuel gas causes the nozzle surface to recede. In addition, the erosion of metal oxide particles ($Al_2O_3$) also plays an important role in ablation [18].

Because of the high-temperature and high-pressure environment in the ablation process, it is difficult to obtain experimental data. With the rapid development of computer technology, numerical simulation through the nozzle model has become another way to study the ablation mechanism. The chemical reaction rate is influenced by both the reaction kinetic rate and the component diffusion rate, and in the fuel gas, $H_2O$ is found is found to contribute the most to the ablation rate [19]. Zhao et al. [20] analyzed the coupling of thermochemical ablation of C/C nozzle and fuel gas, and considered the change of flow field caused by fuel regression during the working condition. For the gas flow containing $Al_2O_3$ particles, its high-speed collision cause erosion and deformation of the nozzle wall. The degree of erosion depends on the properties, the collision speed, and the collision angle of the particle phase [21]. Researchers usually analyze the interaction between the particles and wall based on the two-phase flow field of nozzle, and the mechanical erosion damage model of particles is established to simulate the process of nozzle erosion. Wang and Tian [22] studied the erosion process of C/C nozzle throat insert through a comprehensive method. The results show that with the increase in the metal oxide particles concentration in the gas, the damage of the matrix and the fracture of the exposed fiber tip will intensify. In summary, the ablation process of nozzle involves a series of physical–chemical reactions, and the coupling of the two is very difficult. The common method in engineering design is to estimate the ablation rate based on the wall boundary conditions using as empirical formula, but the empirical parameters vary with different conditions.

The ablation of the nozzle is complex and affected by many factors. For example, the ablation of the nozzle is affected by a combination of temperature, pressure and convective heat transfer coefficient. The empirical formula of nozzle ablation rate can be obtained based on these wall boundary conditions, so the accurate acquisition of parameters in the empirical formula is an important factor to accurately predict the ablation rate.

For a specific problem, obtaining the output parameters directly from the input parameters is a direct problem, which can usually be obtained by experiments or numerical analysis. However, due to the limitations of experimental conditions, experimental methods and theories, some parameters in the problem are usually difficult to obtain directly from experiments, so it is necessary to obtain input parameters through output parameters, which is called the inverse problem. In this paper, the acquisition of parameters in the empirical formula of ablation rate is an inverse problem. The inverse analysis method is to solve the inverse problem.

The inversion analysis method is an indirect method that uses numerical analysis to fit the parameters of the test results [23]. The purpose of the inversion analysis is to make the error between the fitting function and the test result reach the tolerance range by fitting the coefficients continuously. Based on the inversion analysis method, the ablation rate can be empirically fitted by pressure, temperature and surface convective heat transfer coefficient.

In this paper, based on the non-contact 3D laser scanning system, a 3D reconstruction of the ablated nozzle was carried out. Then, the model was cut by the plane of the symmetry axis. Additionally, the contour curve of the cut surface was obtained, which was compared with the design curve to characterize the ablation rate of the nozzle quantitatively. Furthermore, the relationship between the ablation rate and pressure, temperature, and surface convective heat transfer coefficient is investigated. The empirical formula for the ablation rate is obtained empirically by the inversion analysis method. The results show that the non-contact 3D laser scanning system is a valuable method for reconstructing the model of the ablated nozzle. The empirical formula can accurately predict the ablation rate under pressure, temperature, and convection heat transfer coefficient, without complex and expensive tests.

This paper is constructed as follows. In Section 2, the model of the nozzle after ablation is generated by the 3D laser scanning system and the ablation rate of the nozzle is obtained by comparing with the design model of nozzle. In Section 3, the boundary conditions of the nozzle wall are obtained by numerical simulation, and then the empirical formula for predicting the nozzle ablation rate is obtained by the inversion analysis method. The main conclusions are summarized in Section 4.

## 2. Test and Measurement

### 2.1. Physical Model

The nozzle consists of metal shell, throat insert, and thermal insulation. A two-dimensional axisymmetric model of the nozzle is shown in Figure 1. The diameter of the throat is 145 mm, the expansion ratio of the nozzle is 9, and the total length of the nozzle is 553.11 mm. The nozzle works in the mass flow of temperature 3500K and pressure 7.4 MPa for working time 46.8 s.

### 2.2. 3D Reconstruction

In order to measure the ablation rate of the nozzle, a non-contact 3D laser scanning system is established. The nozzle scanning process is shown in Figure 2. The 3D laser scanning system consists of two optical cameras, a light source, and a fringe pattern projector, as shown in Figure 2b. The scanning accuracy of this system is 0.045 mm + 0.3 mm/m, and the scanning speed is 3,000,000 points per second. 3D structural data is obtained by projecting laser gratings onto the nozzle surface. To increase the spatial resolution, the nozzle surface is covered with random marking points, as shown in Figure 2a, and multiple overlapping scans are performed. Finally, the ablated nozzle model is reconstructed by the

3D laser scanning system, as shown in Figure 2c. The 3D laser scanning system used in this test is EinScan Pro 2X 2020, and its manufacturer is SHING 3D.

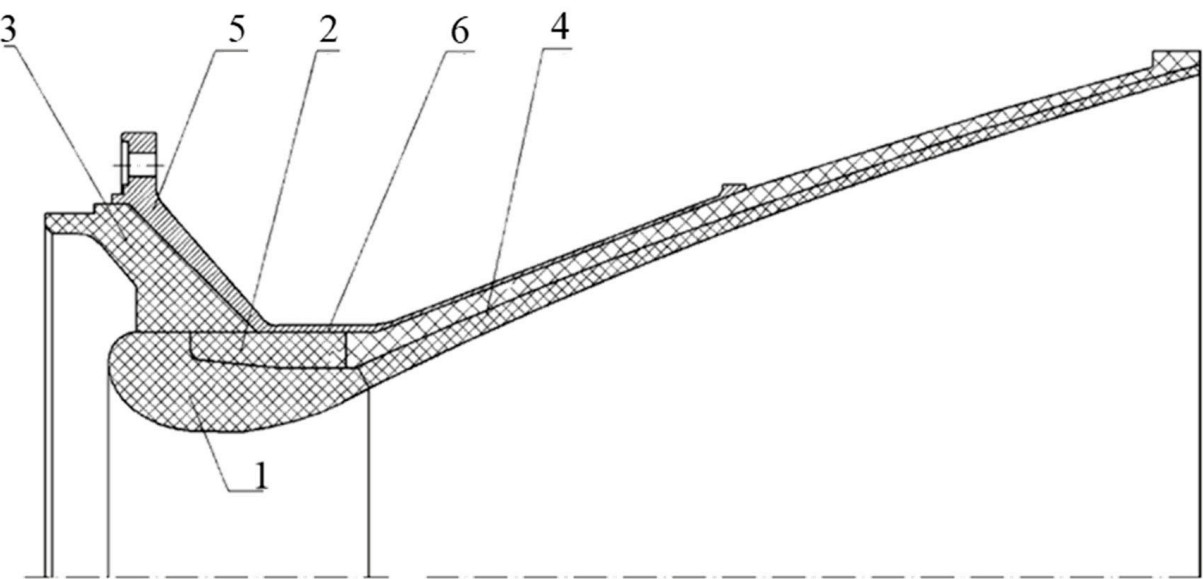

**Figure 1.** Two-dimensional axisymmetric model of the nozzle. 1. Throat insert, 2. Thermal insulation in back wall, 3. Thermal insulation in convergent section, 4. Thermal insulation in divergent section, 5. Shell of convergent section, 6. Shell of divergent section.

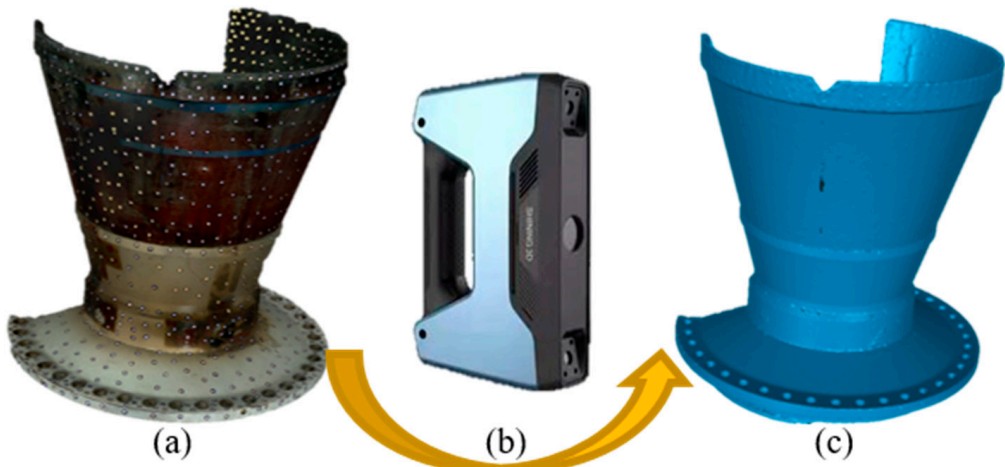

**Figure 2.** The scanning process of the nozzle model. (**a**) Ablated nozzle; (**b**) 3D laser scanning system; (**c**) 3D Reconstruction of Ablated Nozzle Model.

The ablation rate of the nozzle can be obtained by comparing the ablated model with the design model. Firstly, a modified coordinate system is established based on the reverse engineering of CATIA. Additionally, then, the cutting surface is acquired by cutting the model through the symmetry axis. The contour curve of the cut surface is compared with the design curve to quantitatively characterize the ablation rate of the nozzle, as shown in Figure 3. Three contour curves were taken at different positions to reduce the influence of errors, as shown in Figure 3b.

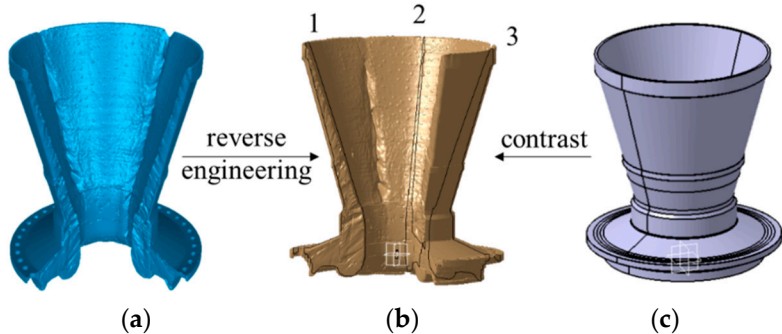

**Figure 3.** 3D reconstruction based on reverse engineering and comparison. (**a**) 3D Reconstruction of Ablated Nozzle Model; (**b**) The modified coordinate system model obtained by reverse engineering; (**c**) Design model of nozzle before ablation.

### 2.3. Analysis of Test Results

The ablation rate, which is equal to the ratio of ablation amount to ablation time, represents the mass loss rate of nozzle material. The ablation rate of the nozzle was obtained by comparing the contour curve of the ablated nozzle with the design curve. Three contour curves at different positions and their average values are brought to reduce the influence of errors, as shown in Figure 4. Three curves marked with "*", "o", and "×" represent the contour curves of the ablated nozzle at position 1, 2, and 3 in the modified model in Figure 3b. The green solid line is the average curve of the three contour curves, and the black dotted line is the design curve before ablation.

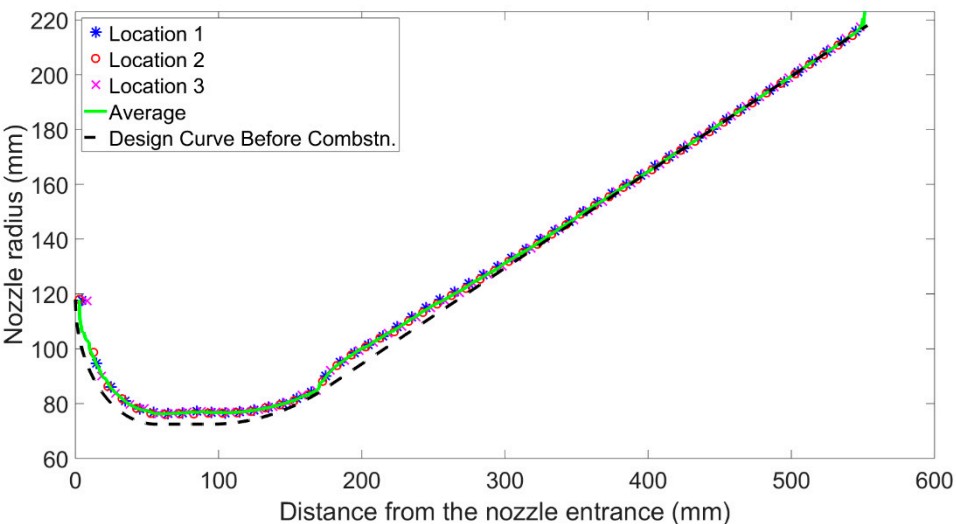

**Figure 4.** Comparison of contour curves of ablated nozzle and design curve.

To quantitatively characterize the ablation rate of the nozzle, $\dot{\delta}$ is defined as follows:

$$\dot{\delta} = \frac{\Delta d}{t} \tag{1}$$

where $\dot{\delta}$ is the ablation rate, $\Delta d$ is the ablation thickness, and $t$ is the working time of the SRM. In this test, the working time $t$ of the SRM is 46.8 s. The ablation thickness $\Delta d$ is the distance from the design curve to the contour curves of ablated nozzle along the normal direction, as shown in Figure 5, so the ablation thickness $\Delta d$ can be defined as follows:

$$\Delta d = \sqrt{\left(y_{aft} - y_{bfr}\right)^2 + \left(x_{aft} - x_{bfr}\right)^2} \tag{2}$$

where $\left(x_{bfr}, y_{bfr}\right)$ is the coordinate on the design curve, $\left(x_{aft}, y_{aft}\right)$ is the corresponding coordinate on the contour curves.

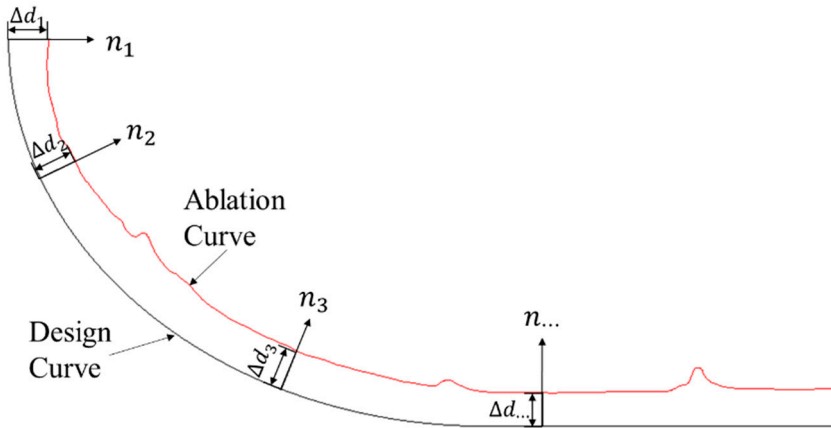

**Figure 5.** Schematic diagram of the amount of ablation $\Delta d$.

The ablation rates at different positions of the nozzle can be obtained by Equations (1) and (2). Figure 6 shows the contour curve of the ablated nozzle, the design curve, and ablation rates at different locations. The red and blue curves are the contour curves of the ablated nozzle and the design curve, respectively. Additionally, the ordinate is on the left side of figures. The black dotted line is the ablation rate, with ordinate on the right side.

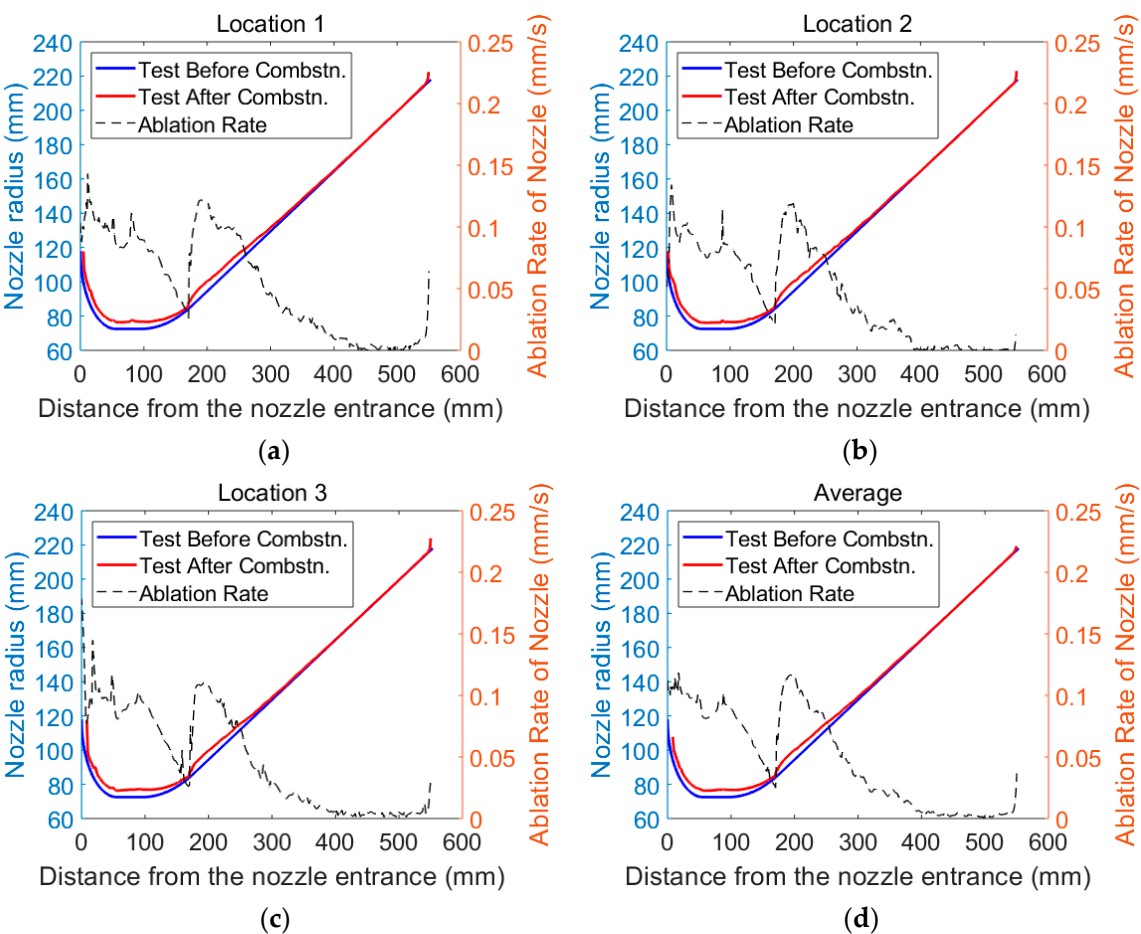

**Figure 6.** Contour curves of ablated nozzle, design curve and ablation rates. (**a**) At position 1; (**b**) At position 2; (**c**) At position 3; (**d**) Average of three positions.

It can be seen from Figure 6 that the ablation rate of the nozzle at the inlet is the most serious, and there is almost no ablation at the end. From the inlet to the end of the nozzle, the ablation rate generally shows a trend from severe to slight. Note that the ablation rate increases sharply at a distance of 200 mm from the nozzle inlet. The throat insert and divergent section are connected here. That is, here is the interface between the two materials.

The sharp increase in ablation rate at the interface between throat insert and divergent section is mainly caused by the different materials of the two. Relatively speaking, the material of the throat insert has better ablation resistance. So, the ablation rate of the divergent section at the interface is greater under the same wall boundary conditions. This results in discontinuity of the nozzle surface at the interface, and the distribution of flow field near the interface also changes: the vortex appears. The appearance of vortex will aggravate the wall boundary conditions at the interface. As a result, the gap of ablation rate between the throat insert and divergent section becomes larger, resulting in a sharp increase in the ablation rate at the interface between the throat insert and divergent section.

## 3. Prediction of Ablation Rate

### 3.1. Numerical Simulation

The ablation of the nozzle, affected by many factors, can cause a pressure drop in the combustion chamber and insufficient combustion. As a result, the prediction of the nozzle ablation rate is crucial. A simulation analysis is carried out to obtain the factors affecting the ablation rate.

The fluid domain calculation model is established, as shown in Figure 7. Additionally, the adiabatic interface condition is applied to the nozzle wall. The temperature of mass flow at the inlet is 3500K, the initial pressure is 7.4 MPa, and the mass flow rate is 75 kg/s. The temperature outside the nozzle is 300K, and the initial pressure is 0.10305 MPa. The model contains 88,535 cells. Additionally, the calculation time is 46.8 s. The simulation results of pressure, temperature, and surface convective heat transfer coefficient are shown in Figure 8.

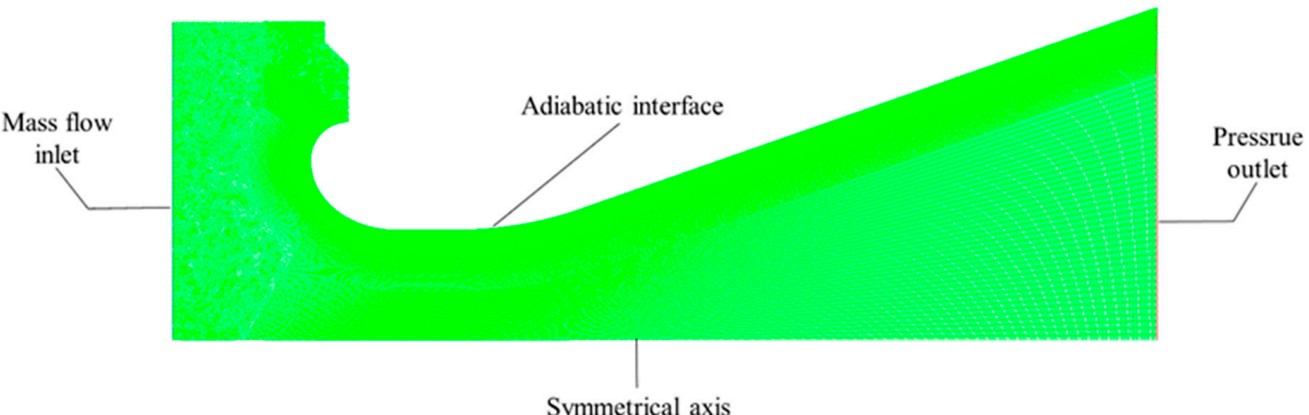

**Figure 7.** Fluid domain calculation model.

The pressure, temperature, and surface convection heat transfer coefficient at the throat insert and divergent section of the nozzle are extracted from the simulation results and compared with the ablation rate obtained in the test. Due to the different materials of the throat insert and divergent section, these two parts are analyzed separately.

The solid lines in Figure 9 are the pressure, temperature, and surface convective heat transfer coefficient at the throat insert, respectively, and the dotted line is the ablation rate at the throat insert. Figure 10 shows the pressure, temperature, convection heat transfer coefficient, and ablation rate at the divergent section. It can be seen from the figure that the ablation rate has the same trend as the pressure, temperature, and convection heat

transfer coefficient in the throat insert and the divergent section. In conclusion, the higher the pressure, temperature, and surface convective heat transfer coefficient are, the greater the ablation rate is.

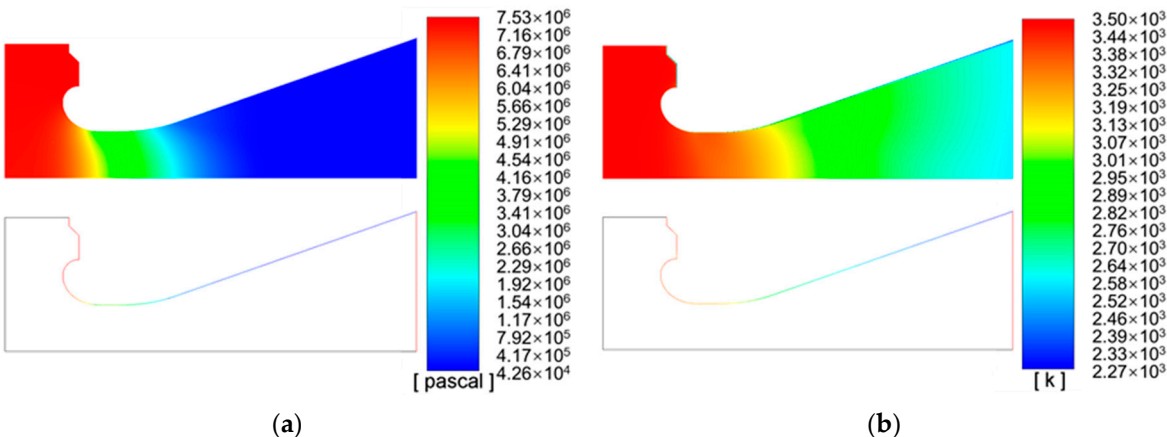

(**a**)                                                                                (**b**)

**Figure 8.** Global and wall simulation results. (**a**) Pressure; (**b**) Temperature.

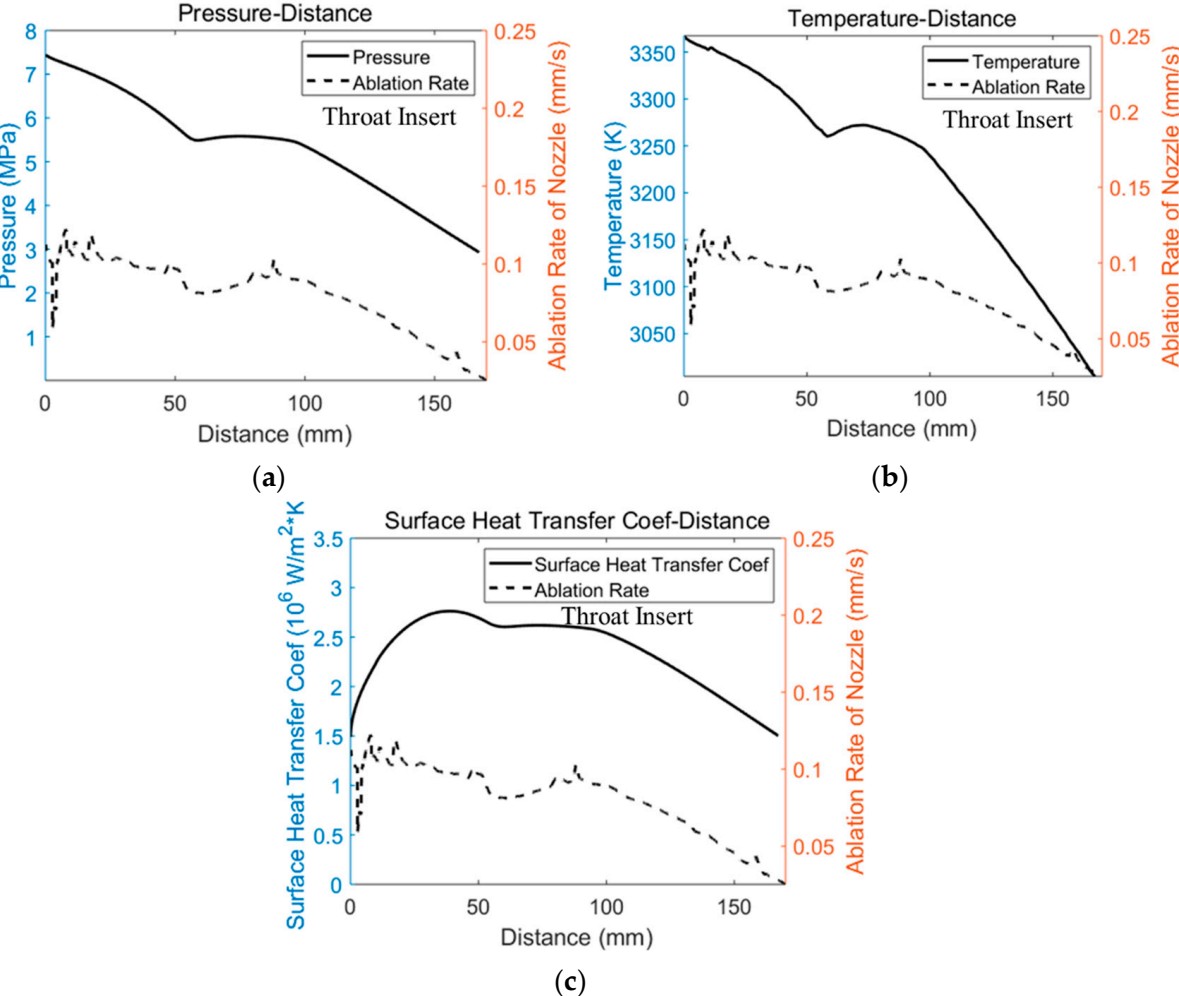

**Figure 9.** The comparison between numerical simulation results and ablation rate at throat insert. (**a**) The pressure and ablation rate; (**b**) The temperature and ablation rate; (**c**) The surface convective heat transfer coefficient.

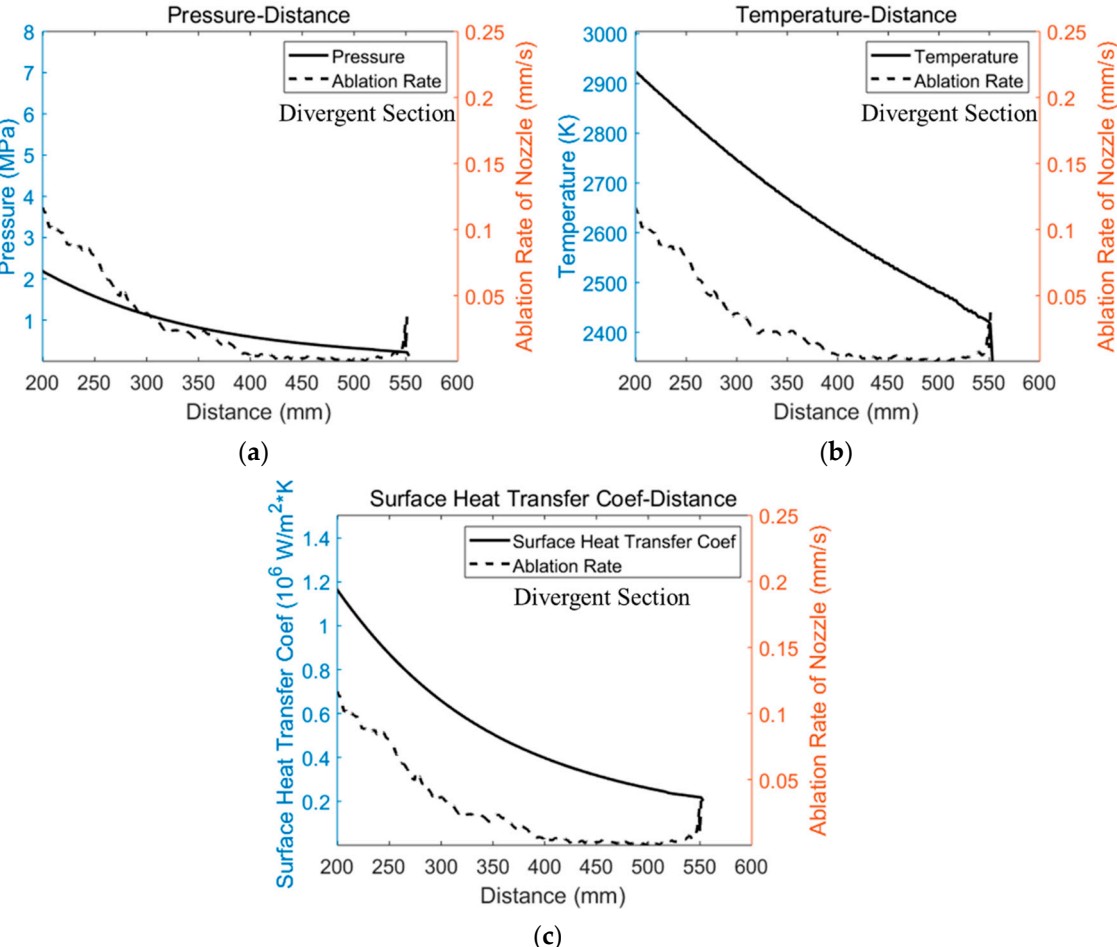

**Figure 10.** The comparison between numerical simulation results and ablation rate at divergent section. (**a**) The pressure and ablation rate; (**b**) The temperature and ablation rate; (**c**) The surface convective heat transfer coefficient.

There is a huge literature since the 60 s that reports the relation between ablation rate and wall boundary conditions. The figures in the literature [24–27] clearly shows that the ablation rate follows the same trend as the pressure, the temperature, and the surface convective heat transfer coefficient all along the nozzle wall indicating a direct correlation between ablation rate and wall boundary conditions. Compared with Figures 9 and 10 in this paper, the accuracy and accuracy of the ablation rate obtained in this paper are also verified. Thus, the feasibility of measuring the nozzle ablation rate by 3D laser scanning system is proved.

*3.2. Inverse Fitting Method*

The analysis shows that the pressure, temperature, and surface convective heat transfer coefficient strongly correlate with the ablation rate, so the empirical formula of ablation rate is obtained empirically using the inversion analysis method based on these parameters. The empirical formula for predicting ablation rate is shown in Equation (3).

$$\dot{\delta}_{fit} = \dot{\delta}_0 + \alpha_1 P^{n_1} + \alpha_2 T^{n_2} + \alpha_3 h_c^{n_3} \tag{3}$$

where $\dot{\delta}_{fit}$ is the fitting data of nozzle ablation rate; $\dot{\delta}_0$ is the basic ablation rate of the throat insert or divergent section; $P$ is the value of pressure divided by $10^6$ MPa, which makes the pressure dimensionless; $T$ is the value of temperature divided by $10^3$ K, which makes the temperature dimensionless; $h_c$ is the value of the surface convective heat transfer coefficient

divided by $10^6$ W/m$^2$·K, which makes the surface convective heat transfer coefficient dimensionless. $\alpha_1$, $\alpha_2$ and $\alpha_3$ are the coefficients of $P$, $T$ and $h_c$; $n_1$, $n_2$ and $n_3$ are the indices of $P$, $T$ and $h_c$.

Based on the inversion analysis method, the ablation rate can be empirically fitted, as shown in Figure 11. The error function can be obtained by comparing the test ablation rate with empirical formula for ablation rate. If the error function does not meet the tolerance range, the coefficient of the empirical formula is updated through the optimization algorithm. Otherwise, if the error function reaches the tolerance range, the inversion analysis method is terminated. The optimization algorithm can be divided into global and local optimization algorithms. The global optimization algorithm is to find the minimum value of the error function in the global range. However, the iterative process of the global optimization algorithm is very complex, resulting in high computational costs [28]. The local optimization algorithm is to find a pretty small value within the tolerance range of the error function [29]. This algorithm has a fast convergence speed and greatly improves the calculation efficiency, so the pattern search method, a local optimization algorithm, is suitable to solve the problem in this paper.

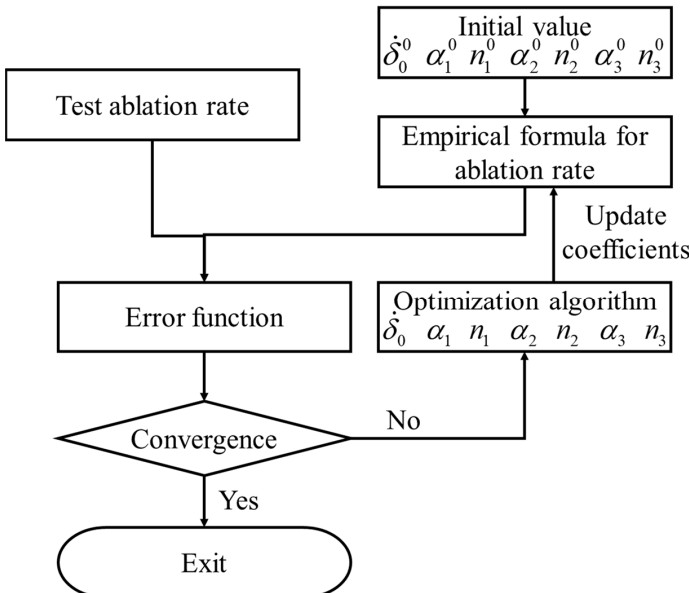

**Figure 11.** Flow chart of inversion identification method.

The pattern search method, also known as Hooke–Jeeves algorithm [30], includes two movement modes: detection movement and pattern movement. The detection movement moves in different directions, detecting the direction that makes the error function decrease. Additionally, the pattern movement moves in this optimal direction at an accelerated speed. In each iteration, the detection movement and the pattern movement are alternated until the algorithm converges.

### 3.3. Fitting Results

The empirical formula for predicting ablation rate is fitted by the pattern search method. The nozzle is divided into a throat insert and a divergent section to fit the empirical formula. The values of basic ablation rate, coefficient, and index of the throat insert, and divergent section are Equations (4) and (5), respectively.

$$
\begin{aligned}
\text{Throat Insert}: \quad & \dot{\delta}_0^t = 0.00101563; \\
\alpha_1^t = 0.011796875; \quad & n_1^t = 1.03710938; \\
\alpha_2^t = 0.000019011; \quad & n_2^t = 0.01171875; \\
\alpha_3^t = 0.002343750; \quad & n_3^t = 2.03125000.
\end{aligned}
\tag{4}
$$

$$\text{Divergent Section}: \dot{\delta}_0^d = 0.00125000;$$
$$\alpha_1^d = 0.023281250; \quad n_1^d = 2.00000000;$$
$$\alpha_2^d = 0.000018841; \quad n_2^d = 0.01562500;$$
$$\alpha_3^d = 0.009785156; \quad n_3^d = 2.37500000.$$

$$(5)$$

The empirical formula curve and test curve of ablation rate are shown in Figure 12. The red line with "×" mark represents empirical formula curve, and the black solid line represents the test curve.

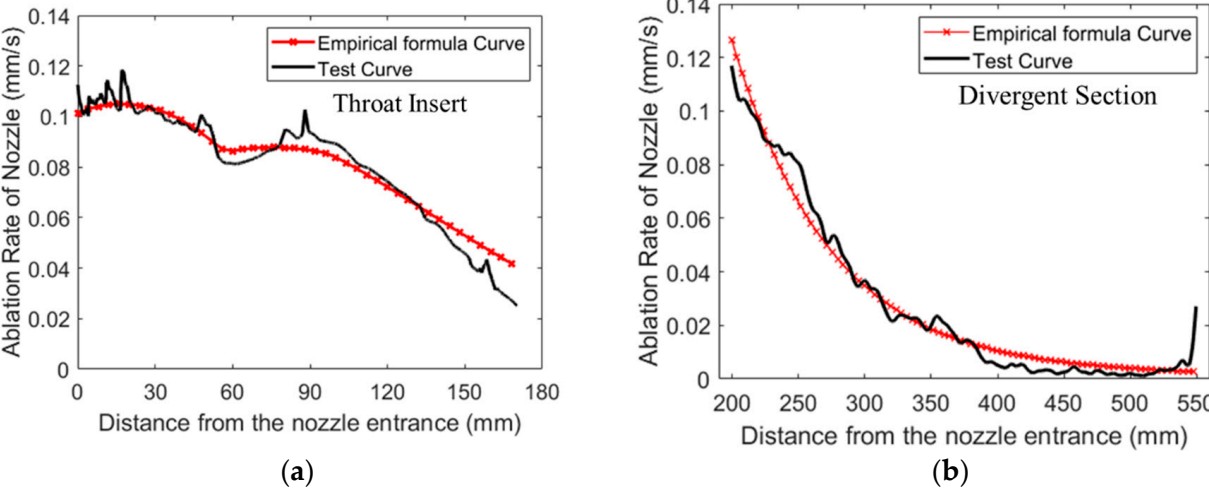

(**a**)  (**b**)

**Figure 12.** Empirical formula curve and test curve of ablation rate. (**a**) At throat insert; (**b**) At divergent section.

According to the empirical formula for predicting ablation rate, the predicted contour curve after ablation can be calculated by Equations (6) and (7).

$$\begin{cases} x_{fit} = x_{bfr} + \sqrt{\dfrac{\left(\dot{\delta} \cdot t\right)^2}{k_n^2 + 1}} \\[4mm] y_{fit} = y_{bfr} + \sqrt{\dfrac{\left(\dot{\delta} \cdot t\right)^2}{\dfrac{1}{k_n^2} + 1}} \end{cases} \tag{6}$$

$$k_n = \frac{y_{aft} - y_{bfr}}{x_{aft} - x_{bfr}} \tag{7}$$

where $\left(x_{fit}, y_{fit}\right)$ is the coordinate of the predicted contour curve after ablation, and $k_n$ is the slope of the normal.

The contrast between the predicted contour curve and test contour curve after ablation is shown in Figure 13. The red line with "×" mark represents predicted contour curve, and the black solid line represents the test contour curve.

From Figures 12 and 13, it can be seen that the overall trend of the empirical formula curve and the test curve of the ablation rate is consistent. The predicted contour curve based on the empirical formula coincided with the test contour curve after ablation of the nozzle, which verifies the accuracy of the empirical formula.

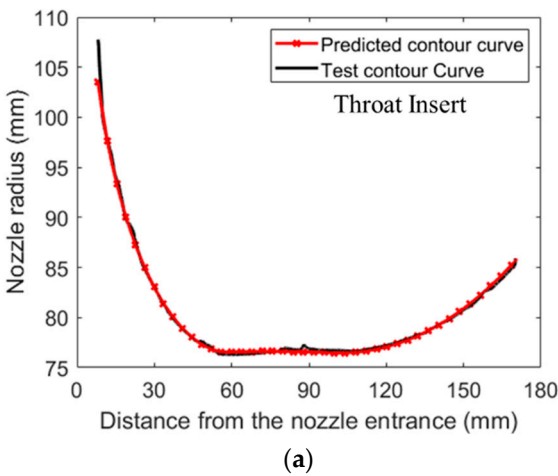 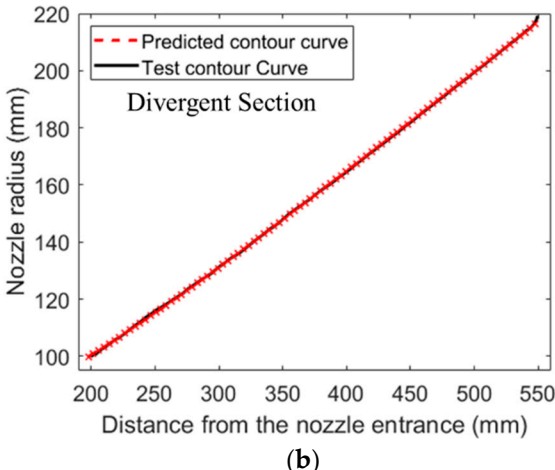

(**a**)                                                                                      (**b**)

**Figure 13.** Predicted contour curve and test contour curve after ablation. (**a**) At throat insert; (**b**) At divergent section.

In conclusion, based on the pressure, temperature, and convection heat transfer coefficient of the numerical simulation, the empirical formula for predicting the ablation rate at the throat insert and divergent section is obtained. The empirical formula can accurately predict the ablation rate under any pressure, any temperature, and any convection heat transfer coefficient, without complex and expensive tests. Additionally, this empirical formula can provide theoretical guidance for nozzle size design and optimization.

## 4. Conclusions

In present work, the ablation rate of the nozzle is quantitatively characterized and measured based on the non-contact 3D laser scanning system. The main conclusions are as follows:

1.  A high-precision model can be obtained by using the non-contact 3D laser scanning system to reconstruct the ablated nozzle. Compared with traditional methods, this method is simpler, more economical, larger scanning size, and can obtain a complete 3D model of ablated nozzle. This 3D model makes it unnecessary to cut the actual specimen for the analysis.
2.  From the inlet to the end of the nozzle, the ablation rate generally shows a trend from severe to slight, which is basically consistent with the changes in pressure, temperature and surface convective heat transfer coefficient obtained from numerical simulation. The conclusion is the same as that of the references, so the accuracy of the ablation rate measured in the paper is verified.
3.  The empirical formula can accurately predict the ablation rate under pressure, temperature, and convection heat transfer coefficient, without complex and expensive tests. Additionally, this empirical formula can provide theoretical guidance for nozzle size design and optimization.

**Author Contributions:** Conceptualization, K.Z., C.W. and Q.L.; methodology, C.W. and Q.L.; software, K.Z. and Z.W.; validation, K.Z. and C.W.; formal analysis, K.Z. and C.W.; investigation, Q.L.; resources, Q.L. and C.W.; data curation, K.Z.; writing—original draft preparation, K.Z. and Z.W.; writing—review and editing, K.Z., C.W. and Q.L.; visualization, K.Z.; supervision, Q.L.; project administration, C.W.; funding acquisition, C.W. All authors have read and agreed to the published version of the manuscript.

**Funding:** This work was supported by the Application Innovation Plan Project of China Aerospace Science and Technology Group (No. 6230112002), the Basic Research Project (No. 514010304-302-2), and the National Natural Science Foundation of China (No. 11772245).

**Institutional Review Board Statement:** Not applicable.

**Informed Consent Statement:** Not applicable.

**Data Availability Statement:** The data presented in this study are available upon request from the corresponding author.

**Conflicts of Interest:** The authors declare that there is no conflict of interest regarding the publication of this paper.

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
