# Peer review of "Characterization of the Nozzle Ablation Rate Based on 3D Laser Scanning System"

_aerospace, doi:10.3390/aerospace10020172_

Round 1

Reviewer 1 Report

The paper is well written and shows a novel method to evaluate the erosion of a nozzle of a solid rocket motor. It deserves to be published with a minor revision.

1) First of all, the authors have to pay attention on the construction of the sentences, as an example in line 66 on page 2: “[…] the prediction […] is crucial because the ablation […]”. In other parts of the manuscript is possible to find a “dot (.)” that interrupts a phrase in a wrong way.

2) The authors are invited to report the model and manufacturer of the 3D apparatus they used in test and measurement section.

3) The authors introduced Kn (the slope of the normal) on page 5. Is there a reason they have introduced this parameter? Is it useful to predict other properties of the materials?

4) The authors have to improve the description of figure 6 explaining better the concepts reported on lines 164-167 especially the concept of “vortex”.

5) In figure 10, figures (b) partially cover figures (a).  

6) On page 7, lines 193-195, the authors stated that “the higher the pressure […], greater the ablation rate”. This is an obvious evidence that does not need a laser technique or a simulation to be proved. There is a huge literature since the 60s that reports it. In order to improve the quality of the paper, the authors should extend the concept to other works comparing their results with others found in literature.  

Author Response

Dear Reviewer:

Manuscript ID: aerospace-2190900

Title: Characterization of the Nozzle Ablation Rate based on 3D Laser

Scanning System

Authors: Kaining Zhang, Chunguang Wang *, Qun Li, Zhihong Wang

Received: 14 January 2023

E-mails: [email protected], [email protected],

[email protected], [email protected]

We greatly appreciate your thorough and thoughtful comments provided on our submitted article. We made sure that each comments have been addressed carefully and the paper is revised accordingly.

The details of the revisions to the manuscript and our responses to the comments are shown in the appendix. Please see below, in blue, for a point-by-point response to the reviewers’ comments and concerns. At the same time, the response to each reviewer are uploaded to the attachment.

Sincerely yours

Dr. Chunguang Wang

Xi’an Jiaotong University, 710049, P.R. China

Appendix

Responds to the Reviewer 1:

The paper is well written and shows a novel method to evaluate the erosion of a nozzle of a solid rocket motor. It deserves to be published with a minor revision.

Question 1:

First of all, the authors have to pay attention on the construction of the sentences, as an example in line 66 on page 2: “[…] the prediction […] is crucial because the ablation […]”. In other parts of the manuscript is possible to find a “dot (.)” that interrupts a phrase in a wrong way.

Response 1:

Thank you for all your comments about our work. We appreciate your professionalism and seriousness. The language problems have been corrected, and we have reviewed the full text.

Question 2:

The authors are invited to report the model and manufacturer of the 3D apparatus they used in test and measurement section.

Response 2:

We are sorry for the unclear description.The 3D laser scanning system used in this test is EinScan Pro 2X 2020, and its manufacturer is SHING 3D. Relevant description has been added to the paper.

Question 3:

The authors introduced Kn (the slope of the normal) on page 5. Is there a reason they have introduced this parameter? Is it useful to predict other properties of the materials?

Response 3:

Thanks a lot for your questions. In Sec.3.3, kn (the slope of the normal) and Equation 7(or Equation 6 in the new manuscript) are used to obtain the predicted contour curve after ablation. The introduction of kn on page 5 is inappropriate, so we adjust its position to Sec.3.3 in the new manuscript.

Question 4:

The authors have to improve the description of figure 6 explaining better the concepts reported on lines 164-167 especially the concept of “vortex”.

Response 4:

We agree with your comments and thank you for your suggestions. We have improved the description of Figure 6. For the reason of the sharp increase of ablation rate, we have made a more detailed description in the manuscript:

The sharp increase of ablation rate at the interface between throat insert and diver-gent section is mainly caused by the different materials of the two. Relatively speaking, the material of throat insert has better ablation resistance. Therefore, under the same wall boundary conditions, the ablation rate of the divergent section at the interface is greater. This results in discontinuity of nozzle surface at the interface, and the distribution of flow field near the interface also changes: the vortex appears. The appearance of vortex will ag-gravate the wall boundary conditions at the interface. As a result, the gap of ablation rate between the throat insert and divergent section becomes larger, resulting in a sharp in-crease in the ablation rate at the interface between throat insert and divergent section.

Question 5:

In figure 10, figures (b) partially cover figures (a).

Response 5:

We are sorry for the unclear figures. We have modified the position of the figures. If it's still difficult to read, we'll deal with it further.

Question 6:

On page 7, lines 193-195, the authors stated that “the higher the pressure […], greater the ablation rate”. This is an obvious evidence that does not need a laser technique or a simulation to be proved. There is a huge literature since the 60s that reports it. In order to improve the quality of the paper, the authors should extend the concept to other works comparing their results with others found in literature.

Response 6:

Your comments are valuable, and we appreciate your suggestions very much. We have added relevant literature and compared our results with those in the literature. In this way, the accuracy and accuracy of the ablation rate obtained in this paper are also verified. Thus, the feasibility of measuring the nozzle ablation rate by 3D laser scanning system is proved.

Reviewer 2 Report

The topic is quite essential for rocket nozzle characterisation. As a non-destructive test method, this 3D laser scanning system is interesting. The resolution of the system is 0.045 mm, and the error bar is ±0.3 mm (page 3, line 109). However, the ablation rates in the results were all within the error limit of 0.25 mm/s. Although understandably, the facility is in the early stages, perhaps this needs further improvement, or this technique could be utilised for quick scanning and other methods for detailed investigation.

Author Response

Dear Reviewer:

Manuscript ID: aerospace-2190900

Title: Characterization of the Nozzle Ablation Rate based on 3D Laser

Scanning System

Authors: Kaining Zhang, Chunguang Wang *, Qun Li, Zhihong Wang

Received: 14 January 2023

E-mails: [email protected], [email protected],

[email protected], [email protected]

We greatly appreciate your thorough and thoughtful comments provided on our submitted article. We made sure that each comments have been addressed carefully and the paper is revised accordingly.

The details of the revisions to the manuscript and our responses to the comments are shown in the appendix. Please see below, in blue, for a point-by-point response to the reviewers’ comments and concerns. At the same time, the response to each reviewer are uploaded to the attachment.

Sincerely yours

Dr. Chunguang Wang

Xi’an Jiaotong University, 710049, P.R. China

Appendix

Responds to the Reviewer 2:

The paper is well written and shows a novel method to evaluate the erosion of a nozzle of a solid rocket motor. It deserves to be published with a minor revision.

Question 1:

The topic is quite essential for rocket nozzle characterisation. As a non-destructive test method, this 3D laser scanning system is interesting. The resolution of the system is 0.045 mm, and the error bar is ±0.3 mm (page 3, line 109). However, the ablation rates in the results were all within the error limit of 0.25 mm/s. Although understandably, the facility is in the early stages, perhaps this needs further improvement, or this technique could be utilised for quick scanning and other methods for detailed investigation.

Response 1:

We are sorry for the unclear description in the manuscript. We reconstructed the ablation nozzle model by 3D laser scanning system. The length of the ablation nozzle is 553mm, so the error of 0.3mm is acceptable. Comparing the three-dimensional reconstruction model of the nozzle after ablation with the design model of the nozzle before ablation, we obtain the ablation thickness Δd. The ablation rate   is the ratio of ablation thickness Δd to time t, as shown in Equation (1). In this paper, the working time t of the SRM is 46.8s. According to Equation (1), the maximum ablation rate is 0.25 mm/s. Therefore, the error of 3d laser scanning system cannot be directly compared with the ablation rate. For a more detailed understanding, we have added a description of the working principle of the 3D scanning system to the manuscript.

For a clearer description, we have added a description of the working principle of the 3D scanning system to the manuscript.
